# Multi-scale fusion framework via retinex and transmittance optimization for underwater image enhancement

**Tie Li**◉[☯], **Tianfei Zhou**◉[☯]*

School of Electronic and Information Engineering, Liaoning Technical University, Huludao, China

☯ These authors contributed equally to this work.
* zhoutianfei2022@163.com

**Data Availability Statement:** All relevant data are within the paper.

**Funding:** This work was supported in part by Foundation of Liaoning Province Education Administration under Grant LJ2020JCL007. The

## Abstract

Low contrast, poor color saturation, and turbidity are common phenomena of underwater sensing scene images obtained in highly turbid oceans. To address these problems, we propose an underwater image enhancement method by combining Retinex and transmittance optimized multi-scale fusion framework. Firstly, the grayscale of R, G, and B channels are quantized to enhance the image contrast. Secondly, we utilize the Retinex color constancy to eliminate the negative effects of scene illumination and color distortion. Next, a dual transmittance underwater imaging model is built to estimate the background light, backscattering, and direct component transmittance, resulting in defogged images through an inverse solution. Finally, the three input images and corresponding weight maps are fused in a multi-scale framework to achieve high-quality, sharpened results. According to the experimental results and image quality evaluation index, the method combined multiple advantageous algorithms and improved the visual effect of images efficiently.

## Introduction

Underwater robots play an essential role in exploiting marine resources and studying biological resources around oceanic hydrothermal vents. In the environment perception of the underwater robot, the operator guides the manipulator to locate the target object and select the grasping attitude through underwater video and image. It is, however, difficult to acquire high-quality sensing scene images through machine vision systems. The quality is severely restricted by the complex underwater environments, such as light attenuation, suspended particles, and artificial lights. As a result, the underwater images we obtained often have color distortion, blur, and low contrast, which dramatically affects the robot's subsequent target recognition and detection tasks. Therefore, correcting the color of the degraded image, eliminating the useless information, and improving the reliability of the different processing results of the image are the primary purposes of clearing the underwater image.

In order to improve the detailed information of underwater images, researchers have proposed various image sharpening methods as preprocessing techniques to restore explicit and natural underwater scenes, including underwater image enhancement, restoration based on

funders had no role in study design, data collection and analysis, decision topublish, or preparation of the manuscript.

**Competing interests:** The authors have declared that no competing interests exist.

physical models, and deep learning. Next, we introduce the representative works of the three types of methods according to the progress time.

The image pixels are processed directly in most underwater image enhancement algorithms. High-contrast images usually exhibit the characteristics of rich details and an extended dynamic range. Contrast limited adaptive histogram equalization (CLAHE) [1] algorithm can improve the contrast of underwater images, but it will introduce noise. The purpose of Retinex theory is to remove the influence of illumination light from an image to obtain the reflection attribute of an object. Yang et al. [2] decomposed V-channel via wavelet transform; then employed denoising algorithm with soft threshold and locally adaptive tone mapping to address the high-frequency and low-frequency components, respectively. Hu et al. [3] further improved its parameters and optimized the performance based on the classical multi-scale Retinex (MSR). Zhang et al. [4] applied a Gaussian low-pass filter to the L channel. They processed low-frequency components through optimal equalization threshold strategy of double interval histogram and enhanced high-frequency components with S-shaped function. Alternatively, Huang et al. [5] used the power function enhancement algorithm for visible light to improve the contrast, corrected the brightness of infrared image, and fused the two versions with Laplace transform. Zhuang et al. [6] established a posterior formulation for underwater image enhancement by imposing multi-step gradient priors on reflectance and illuminance. A piecewise and piecewise linear approximation of reflectivity is modelled with the $l_1$ norm and the $l_2$ norm is used to enhance spatial smoothness and spatial linear smoothness over lighting. Clear images are obtained through convergence analysis and optimization.

Jaffe-McGlamery's underwater optical imaging model [7] has been widely applied in underwater image restoration algorithms, as shown in Fig 1. Among them, He et al. [8] proposed the dark channel prior (DCP) theory built upon extensive statistical research, which indicates that there is at least one low-intensity channel in the red, green, and blue color channels in fog-free images. Consequently, the dark channel prior theory is widely used to estimate the background light and transmittance of haze images. However, since the attenuation characteristics of underwater light are different from that of the land environment, this algorithm cannot get superior results when it is directly applied to underwater images. Based on the DCP, Peng et al. [9] combined image blurriness and red light absorption differences to estimate the scene depth map and estimate the background light from the blurred area. This method is more

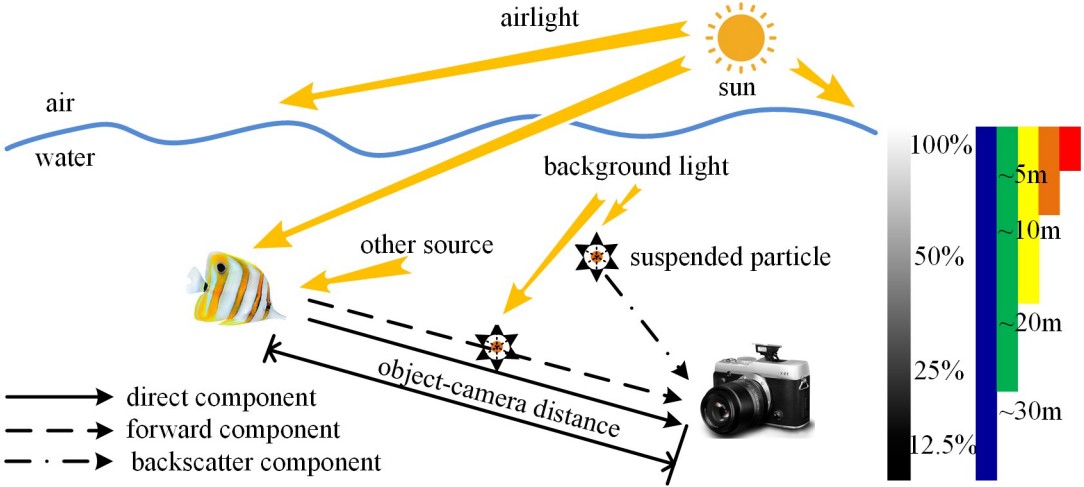

**Fig 1. Underwater optical imaging model.**

suitable for underwater scenes. Drew et al. [10] proposed a priori method of underwater dark channels, which obtained relatively accurate transmittance based on the assumption that most visual information was heavily altered in the blue and green channels. Emberton et al. [11] divided the input image into three categories (bluish, greenish, and bluish green) and restored it according to the color deviation characteristics of each region. Besides, Chang et al. [12] revised the SDCP scheme to make DCP suitable for underwater environments, and adopted point spread function deconvolution strategy to solve the blurred edges.

With the vigorous development of networks, deep learning has been gradually applied to image processing in recent years. Jamadandi et al. [13] utilized the VGG19 network as the encoder and mirrored the wavelet non-pooling layer as the decoder to reconstruct the image model. Lin et al. [14] adopted the deep learning and rectified linear unit (ReLU) activation function to compensate for the linear model through the addition of nonlinear factors and effective feature achievement. Islam et al. [15] proposed FUnIE-GAN based on the U-Net framework. In order to obtain rich feature information, residual connections are added to the generator. To reduce the semantic difference between low-level features and high-level features, Han et al. [16] added residual path blocks between encoders and adopted depth supervision mechanism to improve gradient propagation. Lin et al. [17] proposed a multi-scale deformable convolutional network composed of encoder-decoder, which acquires abundant feature information of the receptive field from different scales, and finally optimized the model through pixel loss and perceptual loss. Li et al. [18] introduced a physical model on the basis of deep learning, and designed a medium-transport guided decoder network to enhance the network's response to quality-degraded regions.

The above methods have selectively improved some issues, such as atomization and color imbalance in underwater images. However, there is a need to deal with detailed information loss, edge contour, and texture blur comprehensively. Some existing fusion methods ignored the selective absorption of water and non-uniform illumination. By simply combining the pure image enhancement algorithms and not fully utilizing the comprehensive value of each advantageous algorithm, local excessive enhancement or color distortion can be seen in the obtained underwater images from other methods. These methods' comprehensiveness, robustness, and accuracy are not ideal and seriously limited in practical application. Considering the degree of image retention, visual effect, and work complexity, we propose a novel underwater optical image enhancement algorithm combining Retinex and transmittance optimization. A clearing method based on image fusion is designed to fuse the dominant information of the image after defogging, contrast enhancement, and color correction. The result is a fog-free, contrast, and color-balanced underwater image.

We analyze the research status of underwater image processing, the shortcomings of existing algorithms, and the idea of this paper in section 1. In what follows, section 2 introduces the related background and main work content of the method. Section 3 conducts experiments on the algorithm proposed in this paper. A detailed and objective evaluation is obtained by comparing with the existing novel algorithms. The last section presents the conclusion of the research.

## Methodology

Generally speaking, our main contributions are summarized as follows. (1) We design three images i, ii, and iii respectively as the input of the fusion framework. i. Histogram quantization is performed on the middle grey area of each channel to improve contrast. ii. Image 2 adopts dynamic adaptive compensation to replace the linear transformation in the color constancy MSR algorithm to correct the image color. iii. Innovatively applies the dual transmittance

imaging model to the underwater image fusion framework and integrates the red channel prior to the total transmittance estimation, which complements the transmittance information and removes the image turbidity. (2) Proposed method extracts the weights of the preprocessed clear input images. After obtaining features and necessary information of the same scene and target, various algorithms are fused to emphasize the image details. (3) Comprehensive and particular verification of the advantages of the algorithm through color correction testing, qualitative and quantitative comparison, complexity analysis, ablation experiments, and application analysis in real underwater. As shown in Fig 2, our study modifies and improves traditional algorithms, resulting in a comprehensive increase in performance for underwater image enhancement.

## Design input images

**Quantify histogram to enhance contrast.** We adjust the image contrast by quantizing the color channel histogram as the first input image. Because of the uniform grey value distribution of the pixels, the three-color channels are divided into the dark, middle gray, and light areas, in which the middle grey area will be quantified. The positive or negative saturation are determined by corresponding to the proportion of gray value pixels. Positive saturation means that pixel numbers at a grey value of 255 exceed 1% of the total pixels. In contrast, negative saturation denotes that pixel numbers at zero grey value covered over 1% of the total ones. The boundary values of the middle grey area are different according to saturation directions. Fig 3 illustrates the upper and lower boundaries of the middle grey area, which represents the normalized histogram values. The orange and green distinguish the dark, bright, and middle grey regions. The boundaries, $V_{min}$ and $V_{max}$, are determined by the black bold lines.

Through the above analysis, the grey value of $[V_{min}, V_{max}]$ is linearly mapped, as shown in Eq (1). Fig 4 shows the initial images, contrast enhancement results, and R, G, and B channel histograms. The contrast has been visibly improved with higher grey level distributions compared to the initial values.

$$\begin{cases} J_{out}(x) = 0.05\dfrac{J}{V_{min}}, J < V_{min} \\[2mm] J_{out}(x) = 0.05 + \dfrac{0.9(J - V_{min})}{V_{max} - V_{min}}, V_{min} \le J \le V_{max} \\[2mm] J_{out}(x) = 0.95 + \dfrac{0.05(J - V_{max})}{1 - V_{max}}, J > V_{max} \end{cases} \tag{1}$$

**Color correction based on Retinex theory.** Underwater images often appear in blue and green tones. To further restore image color, the second input image is generated from the improved multi-scale Retinex with color preservation (IMSRCP) based on the Retinex theory. According to color constancy, the image is represented as the product of scene illumination and object reflection components. The reflection components of the object are obtained by removing the influence of the scene illumination component. We pre-correct and homogenize the color and reduce the blue-green bias. After that, the R, G, B, and L channels are corrected through MSR [3] to enhance the colors under the premise of high fidelity and L is the mean value of channels R, G, and B.

$$R^i_{msr} = \sum_{n=1}^{N} \omega_n(\log(I^i(x, y)) - \log(F(x, y, \sigma_n) * I^i(x, y))) \tag{2}$$

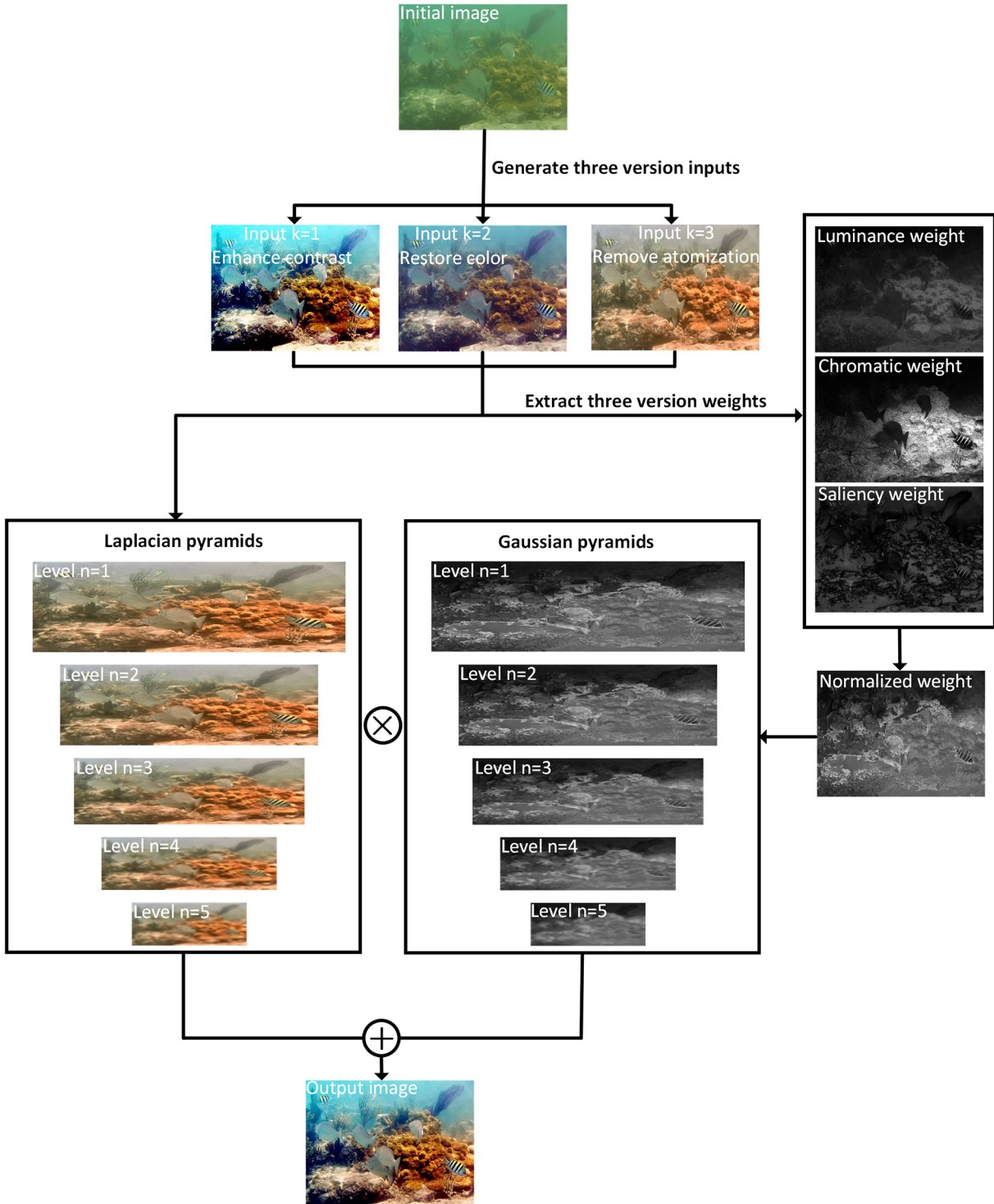

**Fig 2. The three input images including contrast enhancement, color correction, and defogging are decomposed into five-layer Laplacian pyramids, respectively, and its normalized weight map is decomposed into five-layer Gaussian pyramids.** (To simplify the flowchart, we only show the defogging weight map and its pyramid level decomposition in the figure). Finally, multi-scale fusion is performed to obtain the output image.

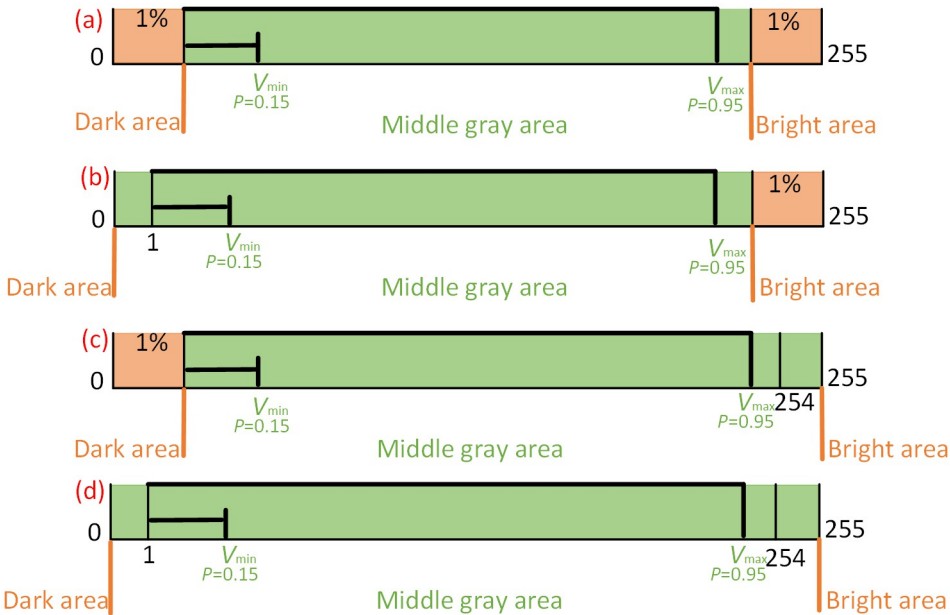

**Fig 3. Upper and lower boundaries of middle gray area.** (a) Unsaturation; (b) Negative saturation; (c) Positive saturation; (d) Negative and positive saturation.

where $\{i \in R, G, B, L\}$, $N$ denotes the scales, $\omega_n$ represents the weights of each scale, $F(x, y, \sigma_n)$ and $\sigma_n$ are Gaussian wrap function and scale, subject to $\iint F(x, y) = 1$.

In the standard MSR algorithm, the gain and offset parameters could not adapt to some specific conversion of different types of images. Therefore, we adopt the dynamic adaptive

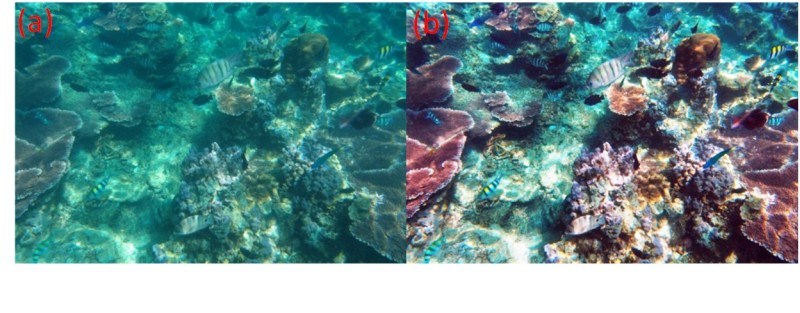

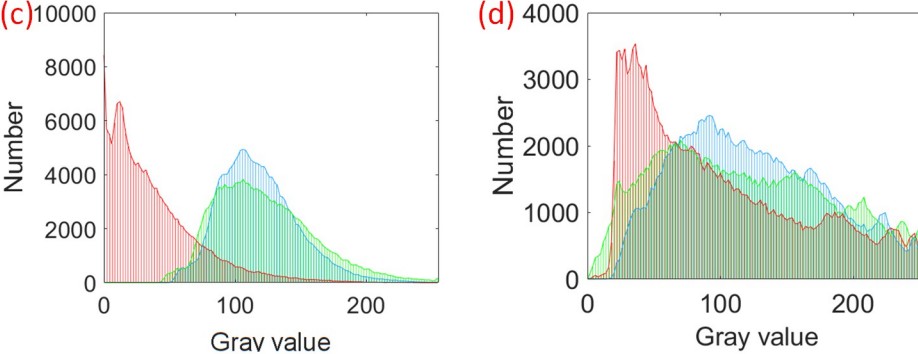

**Fig 4. Contrast image and RGB three-channel histogram.** (a) Original; (b) Contrast enhancement; (c) Original histogram; (d) Histogram after contrast enhancement.

tensile compensation to tune the parameters dynamically and address this problem, given by Eq (3). Besides, our proposed method retains the original necessary color and enhances the color saturation through the method of the anti-gray world in Eq (4).

$$\begin{cases} I^c_{msr_min} = I^c_{msr_mean} - \mu I^c_{msr_var} \\[2mm] I^c_{msr_max} = I^c_{msr_mean} + \mu I^c_{msr_var} \\[2mm] I^c_{imsr} = 255\dfrac{R^c_{msr} - I^c_{min}}{I^c_{max} - I^c_{min}} \end{cases} \tag{3}$$

$$I^c_{imsrcp} = \rho\frac{I^c_{mean}}{I^L_{imsr_mean}}I^c_{imsr} \tag{4}$$

where $\{c \in R, G, B\}$, $\mu$ is the dynamic range, $I_{msr\_mean}$ and $I_{msr\_var}$ refer to the mean and standard deviation of the MSR processed image. $I_{mean}$ demonstrates the mean value of input images, $I_{imsr\_mean}$ refers to the light intensity channel of the improved MSR (IMSR) processed image, and $\rho$ is the color compensation coefficient. Fig 5 compares the experimental results based on the Retinex algorithm, which shows that IMSRCP is less competent in removing turbidity but results in a more distinct color without exposure.

**Dehazing algorithm based on imaging model.**    The contrast and color enhancement images can be obtained from the first and second input images. Then, the turbidity phenomenon is addressed using the third input image. Based on the different attenuation coefficients of direct and backscattering components in the underwater imaging model [19], the algorithm redefines the underwater imaging framework, provided by Eq (5). Compared with traditional models, we add a variable parameter and improve the accuracy of the underwater image restoration algorithm.

$$I_c(x) = J_c(x)t_D(x) + [1 - t_B(x)]A_c \tag{5}$$

where $x$ is the pixel, $I$ and $J$ represent the blurred image taken by the camera and the clear image to be restored. $A_c$ denotes the background light. Besides, we define the direct and backscattered component transmittance $t_D(x)$ and $t_B(x)$ as Eqs (6) and (7):

$$t_D(x) = \exp[-\sigma_D z(x)] \tag{6}$$

$$t_B(x) = \exp[-\sigma_B z(x)] \tag{7}$$

where $\sigma_D$ and $\sigma_B$ are direct and backscatter attenuation coefficients, respectively, related to propagation distance and light wavelength. $z(x)$ is the distance from pixel to camera.

We estimate the background light by the hierarchical search of quadtree to prevent the background value from increasing due to artificial light sources and bright white objects. Here,

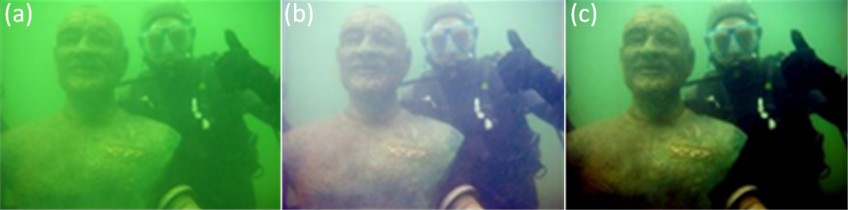

**Fig 5. Different enhancement effects of the algorithms based on Retinex.** (a) Original image; (b) MSR; (c) IMSRCP.

we divide the image into four areas to find the image block with the smallest standard deviation and largest mean to set as the target. The target region is further divided into four smaller ones, repeated until the specific threshold. In the final selected region, we choose the point with the minimal Euclidean distance from the pure white pixel as the background light value.

The weak correlations between the attenuation coefficient of backscattering transmittance and the light wavelength have been ignored in our work, which means the backscattering transmittances of R, G, and B channels are consistent. To correct the color deviation caused by the low R-channel in DCP [7], we apply the red dark channel prior [20]: $J^{RDCP}(x) \approx 0$. Also, we divide both sides of the dual-transmittance underwater imaging model by the background light as follows:

$$\frac{I_c(x)}{A_c} = \frac{J_c(x)}{A_c} t_D(x) + [1 - t_B(x)] \tag{8}$$

In addition, we take the red dark channel prior into Eq (8) to obtain the backscattering transmittance:

$$t_B(x) = 1 - \min\left\{ \frac{\min\limits_{y \in \Omega(x)}[1 - I_R(y)]}{1 - A_R}, \frac{\min\limits_{y \in \Omega(x)}[1 - I_G(y)]}{A_G}, \frac{\min\limits_{y \in \Omega(x)}[1 - I_B(y)]}{A_B} \right\} \tag{9}$$

While taking images in underwater environments, distances between objects that are short enough to have little influence on direct component transmittance, which are ignored within the proposed algorithm. According to the estimation of backscattering transmittance and the relations between the two transmittances, we can further estimate the direct transmittances at all the pixels from Eqs (6) and (7).

$$\frac{\sigma_D}{\sigma_B} = \frac{\ln t_D(x)}{\ln t_B(x)} \tag{10}$$

$$t_D(x) = \{\exp[-\sigma_B z(x)]\}^{\frac{\sigma_D}{\sigma_B}} \tag{11}$$

The restored image is obtained in Eq (12) by integrating the estimations into the original formula 5, where $t_0$ is a threshold to prevent small transmittance, set to 0.3 empirically [21].

$$J_c(x) = \frac{J_c(x) - A_c[1 - t_B(x)]}{\max[t_D(x), t_0]} \tag{12}$$

It can be shown from Fig 6(b) that our work can estimate the locations of the background light value accurately. Besides, represented alongside in Fig 6(c) depicts the backscattering transmittance of the image. The more serious the color deviation is, the more pronounced color change appears after restoration. Through Fig 6(d), 6(e) and 6(f), we can verify that this method conforms to the principle of underwater light propagation, in which red light decays the fastest, followed by blue and green lights. It can be seen from figures (g), (h), (i), and (j) that the transmittance maps can accurately reflect the image depth of field information.

## Design weight diagrams

With the three fused input images, we designed three weight maps representing the basic features, necessary information, and pixels with high weight, including luminance, chromatic, and saliency maps.

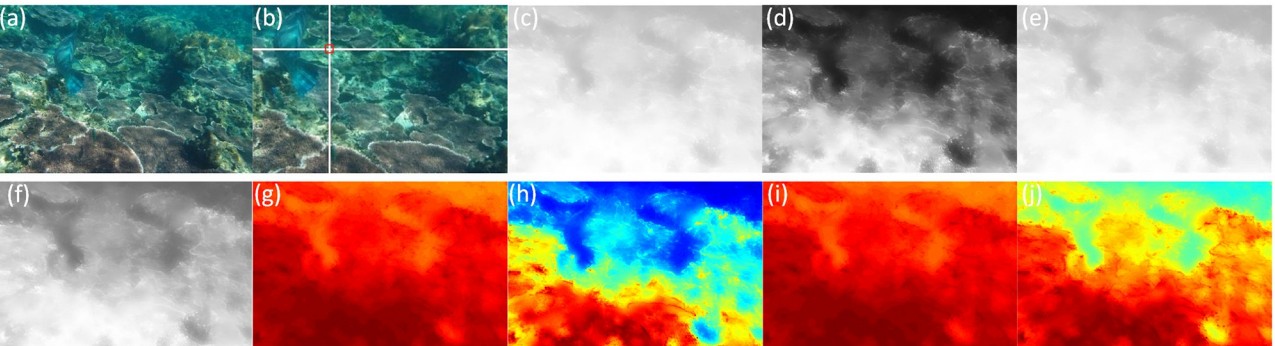

**Fig 6. Image transmission.** (a) Original; (b) Background light; (c) Backscatter component transmission; (d) R channel direct component transmission; (e) G channel direct component transmission; (f) B channel direct component transmission; (g) Thermodynamic diagram of figure (c); (h) Thermodynamic diagram of figure (d); (i) Thermodynamic diagram of figure (e); (j) Thermodynamic diagram of figure (f).

(1) Luminance map $W_L$. The luminance map assigns high saturation values to high visibility areas, resulting in images with higher visibility. It is utilized to reflect the image's brightness, which can effectively measure the lack of brightness. Calculate the average value of three channels of each input image, the standard deviation of the R, G, and B channels and the average value is the luminance map:

$$W_L = \sqrt{1/3[(R-L)^2 + (G-L)^2 + (B-L)^2]} \qquad (13)$$

where R, G, and B are the three-channel pixel values, and L is the average value of the three-channel pixels.

(2) Chromatic map $W_C$. Also, the chromatic map compensates for the disadvantage of color reduction and reflects the color purity and overall quality of the image. Convert the image to HSV space, calculate the saturation of each pixel and the maximum saturation in the region:

$$W_C = \exp\left(-\frac{(S(x) - S_{max})^2}{2\sigma^2}\right) \qquad (14)$$

where $S(x)$ demonstrates the saturation of the pixel $x$, $S_{max}$ represents the maximum saturation, which is taken as 1 in our method, $\sigma$ controls the sensitivity of the standard deviation, and the value of 0.3 is more effective.

(3) Saliency map $W_S$. The saliency map calculates the standard deviation between the brightness of the pixel and the local average of the surroundings to recover more details:

$$W_S = \|I_{mean} - I_{whc}\| \qquad (15)$$

where $I_{mean}$ represents the average value of pixels, and $I_{whc}$ means the luminance channel obtained after low-pass filtering.

(4) Normalized map $\bar{W}$. Normalize the above weight map to get the corresponding standardized weight map:

$$W_k = W_L^k + W_C^k + W_S^k \tag{16}$$

$$\bar{W}_k = \frac{W_k}{\sum\limits_{k=1}^{K} W_k} \tag{17}$$

where $k$ is the serial number of the input image, $\bar{W}_k$ is the normalized weight map of the k-th input image, $K = 3$.

## Multi-scale decomposition and fusion

In order to obtain multi-scale features, the Gaussian and Laplacian pyramids are applied to decompose the input image $I_k$ and normalized weight image $\bar{W}_k$. Image pyramid is a technique for extracting multi-scale features of an image. By sampling the image several times, a group of characteristic images with different scales with reduced resolution and image size are obtained.

(1) Constructing Gaussian pyramid: the Gaussian pyramid is acquired by smoothing the brightness through low-pass filtering and down sampling compression size. Gaussian smoothing brightness filtering is realized by Gaussian function generation kernel. Down sampling is obtained by sampling the image undergone Gaussian smoothing with interlaced rows and columns.

(2) Constructing Laplacian Pyramid: the specific implementation process is to subtract the Gaussian pyramid of the same layer from the interpolation image expanded by interpolation on the previous layer of this layer.

Finally, a multi-scale fusion method is utilized to reconstruct all pyramid layers to receive a clearer output image.

$$R_n(x) = \sum_{k=1}^{K} G_n\{\bar{W}_k(x)\} LP_n\{I_k(x)\} \tag{18}$$

where $n$ and $k$ determine the number of pyramid layers and input images, in this paper, the total number of layers of the pyramid is $N = 5$, $G_n$ represents the $n$ layer of Gaussian decomposition of the normalized weight map, and $LP_n$ represents the $n$ layer of Laplace decomposition.

## Experimental results and analysis

We compare several existing algorithms from different directions to verify the effectiveness and robustness of our method in different water environments, including underwater dark channel prior (UDCP) [10], double red-dark channel prior (DRDCP) in Wang et al. [21] for the dual transmittance imaging model, the Fusion algorithm proposed by Ancuti et al. [22], UNTV is based on a red channel prior guided variational framework [23], multilevel feature fusion-based conditional gan (MLFcGAN) [24], a fast underwater image enhancement for improved visual perception (FUnIE-GAN) [15], underwater image enhancement convolutional neural network(UWCNN) [25] and a deep underwater image enhancement network (Water-Net) [26].

## Color correction test

In order to verify the accuracy of color restoration of proposed algorithm, the above-mentioned multiple algorithms were subjected to underwater color card calibration experiments and compared with the standard color card. Fig 7(a) and 7(b) present the distorted color card images taken in underwater environments and the standard card.

As shown in Fig 7, the image color recovery after UDCP processing is good, but the green and purple color blocks cannot be distinguished. Most of the image color blocks processed by DRDCP fail to restore the correct color depth. After the fusion method, the image has a darker tone overall. The red and blue blocks are more severely distorted, and the discrimination between different degrees of green blocks is low. The UNTV algorithm is seriously distorted. MLFcGAN processing color card is fuzzy, which results in the loss of color information. Color contrast of the algorithm has not been effectively improved in FUnIE-GAN even though its color block is close to the actual color card. Blue turbidity happens in the UWCNN, making it impossible to distinguish between various color systems. The color card processed by DUIENet has the phenomenon that the color information is not apparent. Combining the above algorithms, the color after processing by ours is very close to the standard card, and the contrast is significantly improved.

## Qualitative comparison

To further consider the restoration of texture details, the Canny operator is employed to verify the algorithm. The more edges and sharpness of the texture, the higher the image information quality, as shown in Fig 8.

Observing Fig 8, we can see that the texture information in the red frame has been improved, indicating that the algorithm can effectively restore texture details, remove the fog, and improve image visibility.

In addition, we test different degrees of degraded images obtained in a complex underwater environment to verify the algorithm's effectiveness further, shown in Fig 9.

The degraded images we selected are from the following three datasets.

(1).  UIEBD [26]: They are collected from Google, YouTube, and related papers, containing 950 degraded underwater images.

(2).  EUVP [15]: Seven cameras are used to capture images rich in underwater scenes, including a set of paired and unpaired 20K underwater images (poor quality and good quality), with a total of 515 data.

(3).  RUIE [27]: A system based on underwater multi-view imaging shooting to obtain data of marine life in the Yellow Sea, covering 4230 images. We selected 12 representative underwater images in these three datasets.

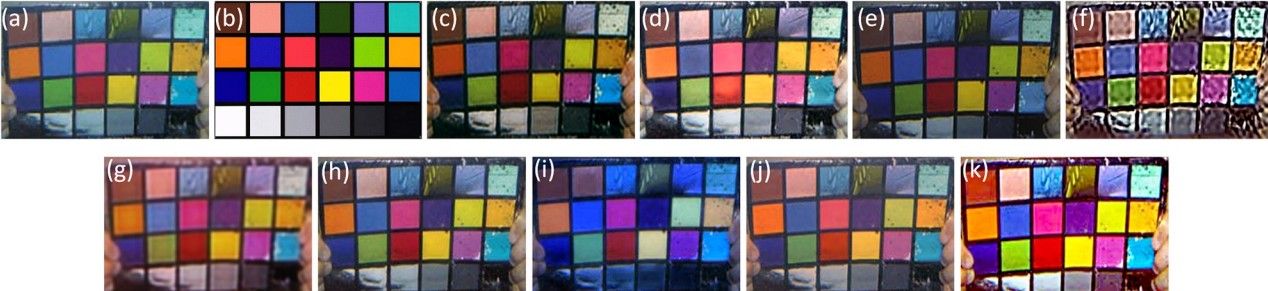

**Fig 7. Color correction test.** (a) Degraded color card; (b) Standard card; (c) UDCP; (d) DRDCP; (e) Fusion; (f) UNTV; (g) MLFcGAN; (h) FUnIE-GAN; (i) UWCNN; (j) Water-Net; (k) Ours.

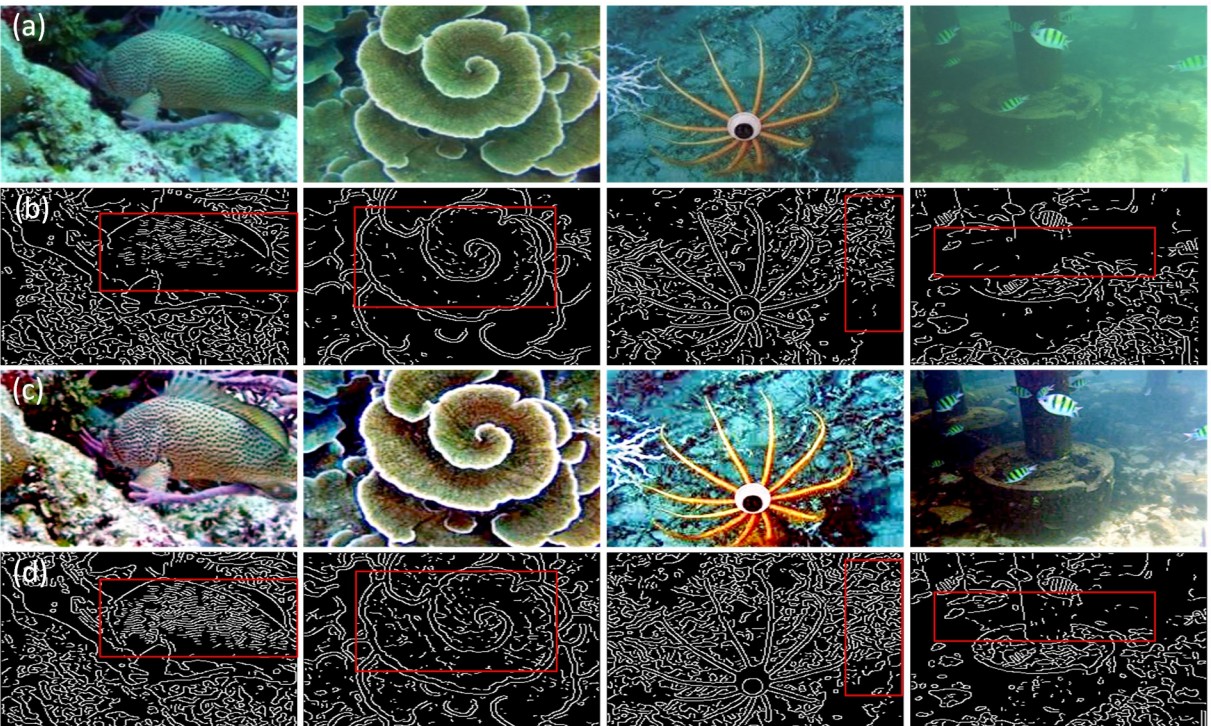

**Fig 8. Canny local texture comparison images.** (a) Degraded image; (b) Degraded canny detection image; (c) Restore image; (d) Recovery canny detection image.

Among them, UIEBD is rich in different types of underwater scenes that we selected seven images in this dataset. Due to the small amount of EUVP data, three representative images are adopted. Although the RUIE dataset contains many images, we only selected two of them because most of the images collected by this dataset are identical kinds of degraded images in a similar region.

As shown in Fig 9, images 1-7 are from UIEBD, 8-10 are from EUVP, and 11 and 12 are from RUIE. Overall, the effect of UDCP and Fusion are poor no matter in which dataset and they deepen the tone of the original degraded image. The color correction effect of DRDCP is better, but its brightness is not improved. UNTV virtualizes the detailed information and adds different degrees of noise. MLFcGAN does not correctly remove the atomization phenomenon of the image, as shown in images 2, 4, and 12. FUnIE-GAN performs better in EUVP but has a poor ability to improve visual effects in UIEBD and EUVP. Similarly, the overall image generated by the UWCNN algorithm is dark blue with serious atomization, resulting in low discernability between color blocks. Water-Net outputs various degrees of red artefacts and fails to restore the image's clarity effectively. However, our method could remove the turbidity and correct the color information naturally with increased contrast.

## Quantitative comparison

To further quantitatively verify the performance of our algorithm, we adopt underwater color image quality evaluation (UCIQE) [28] and underwater image quality measures (UIQM) [29] evaluation indexes.

UCIQE takes chroma, saturation, and brightness as a linear combination and the value of UCIQE is in the range of [0, 1], directly proportional to the underwater image quality. Table 1

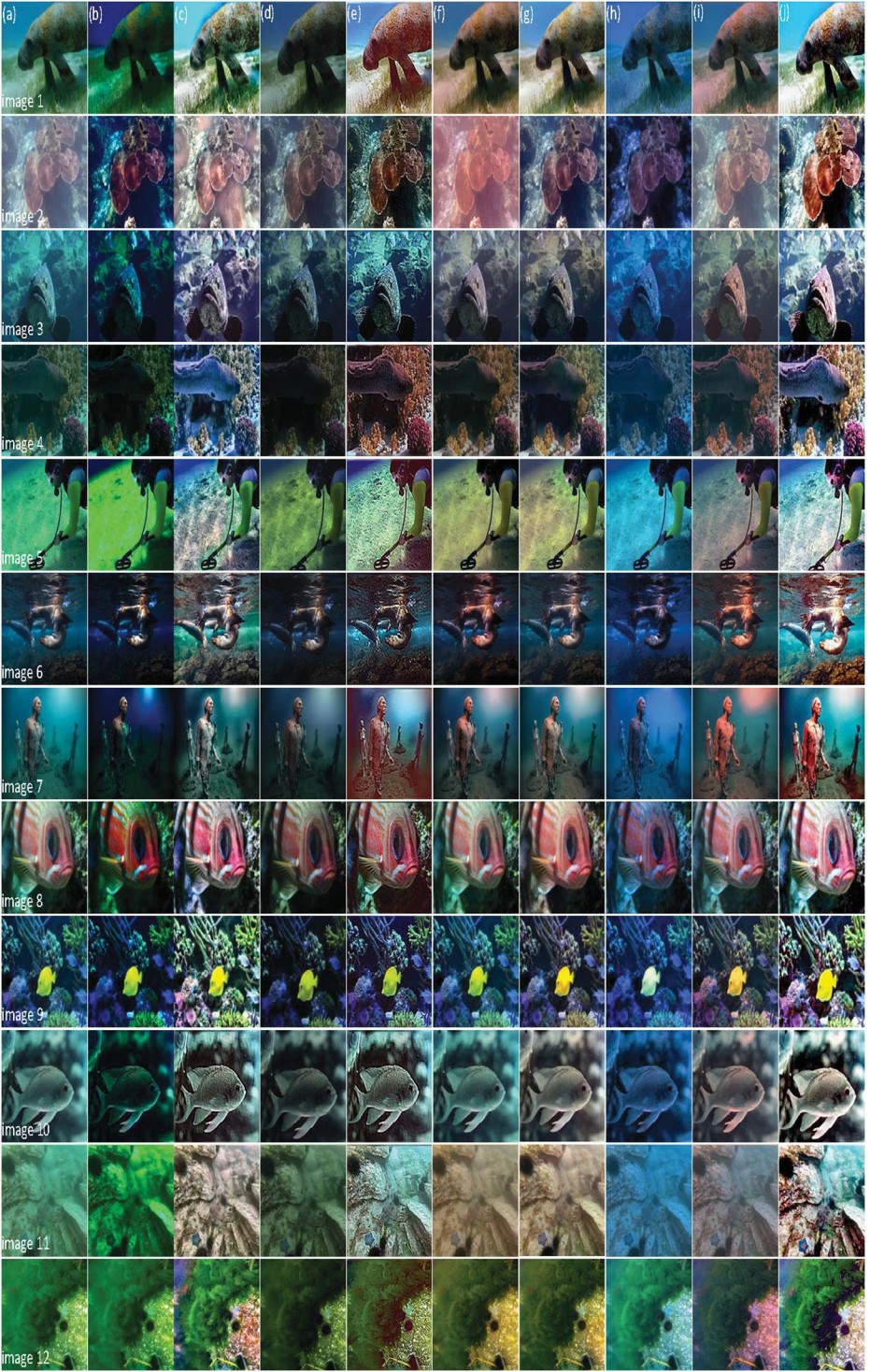

**Fig 9. Experimental results in complex underwater environments.** (a) Degraded image; (b) UDCP; (c) DRDCP; (d) Fusion; (e) UNTV; (f) MLFcGAN; (g) FUnIE-GAN; (h) UWCNN; (i) Water-Net; (j) Ours.

**Table 1. Quantitative evaluation results of UCIQE index.**

| Image | UDCP | DRDCP | Fusion | UNTV | MLFcGAN | FUnIEGAN | UWCNN | WaterNet | Ours |
|-------|------|-------|--------|------|---------|----------|-------|----------|------|
| 1 | 0.5330 | 0.6479 | 0.5245 | 0.6777 | 0.5913 | 0.6238 | 0.4898 | 0.5605 | **0.6829** |
| 2 | 0.6517 | 0.5141 | 0.5154 | 0.6572 | 0.4597 | 0.5838 | 0.5444 | 0.4706 | **0.7038** |
| 3 | 0.5562 | 0.5775 | 0.5163 | 0.6253 | 0.4987 | 0.5438 | 0.5345 | 0.4947 | **0.6268** |
| 4 | 0.5315 | 0.6037 | 0.4624 | 0.6547 | 0.5475 | 0.5870 | 0.4599 | 0.5362 | **0.6706** |
| 5 | 0.6070 | 0.6012 | 0.5329 | 0.6309 | 0.5645 | 0.5825 | 0.5759 | 0.5300 | **0.6336** |
| 6 | 0.5746 | 0.6090 | 0.5154 | 0.6554 | 0.5647 | 0.5692 | 0.5256 | 0.5922 | **0.7079** |
| 7 | 0.5310 | 0.5819 | 0.5337 | 0.6577 | 0.5781 | 0.5957 | 0.4969 | 0.5864 | **0.7034** |
| 8 | **0.6687** | 0.6354 | 0.5636 | 0.6085 | 0.6033 | 0.6326 | 0.5754 | 0.5966 | 0.6545 |
| 9 | 0.6316 | 0.6300 | 0.5802 | 0.6126 | 0.6007 | 0.6197 | 0.6181 | 0.6036 | **0.6632** |
| 10 | 0.5272 | 0.5504 | 0.4875 | 0.6105 | 0.5168 | 0.5174 | 0.5130 | 0.5078 | **0.6125** |
| 11 | 0.5431 | 0.5109 | 0.4437 | 0.5998 | 0.4418 | 0.5164 | 0.4340 | 0.4457 | **0.6519** |
| 12 | 0.5579 | 0.6012 | 0.5500 | 0.6121 | 0.5887 | 0.5971 | 0.5123 | 0.5660 | **0.6416** |

shows the UCIQE evaluation results corresponding to the underwater image clarification algorithms in Fig 9, with the bold font emphasizing the best performance.

UIQM quality evaluation index takes color, sharpness, and contrast as the basis for evaluating image quality. This value is directly proportional to image quality, as shown in Table 2. Similarly, bold font is the optimal value.

More generally, when evaluated with UCIQE and UIQM, the proposed method performs better than others in various underwater surroundings with remarkable robustness. Then, we test the UCIQE and UIQM values of the datasets EUVP, RUIE, and UIEBD, and show their average values in Tables 3 and 4 to prove the universality and robustness of our method.

According to Tables 3 and 4, our algorithm performs better than others in the three datasets and is suitable for various complex underwater environments. In order to show the advantages of the algorithm more intuitively, the above table is drawn as a scatter chart, as shown in Fig 10. It can be seen from the scatter chart that ours is not only the best but also the most stable. When other comparison algorithms perform better on the UIEBD and EUVP datasets, ours can far exceed their index values. When the UCIQE and UIQM values of different comparison algorithms are relatively low in the RUIE dataset, ours still maintains the best results.

**Table 2. Quantitative evaluation results of UIQM index.**

| Image | UDCP | DRDCP | Fusion | UNTV | MLFcGAN | FUnIEGAN | UWCNN | WaterNet | Ours |
|-------|------|-------|--------|------|---------|----------|-------|----------|------|
| 1 | 3.1716 | 3.5773 | 4.5275 | 5.0038 | 4.8088 | 4.7825 | 3.4649 | 4.7716 | **8.1785** |
| 2 | 4.5205 | 5.5292 | 5.2383 | 3.6837 | 5.0588 | 5.5204 | 5.2492 | 4.5950 | **7.1215** |
| 3 | 4.2379 | 5.3473 | 4.5183 | 2.8589 | 4.5040 | 4.9275 | 3.5251 | 4.3934 | **5.3745** |
| 4 | 5.0529 | 3.8639 | 5.2069 | 4.7492 | 5.1663 | 5.2295 | 3.9852 | 5.0823 | **5.6102** |
| 5 | 1.0546 | 3.9628 | 3.9284 | 3.4201 | 4.0649 | 4.7369 | 1.8711 | 4.5993 | **6.8025** |
| 6 | 4.2790 | 4.9226 | 5.3751 | 3.6184 | 5.2401 | 5.1463 | 4.0202 | 5.2799 | **9.0879** |
| 7 | 3.2834 | 3.9030 | 4.0136 | 4.6684 | 3.8025 | 3.8648 | 2.2851 | 4.1149 | **5.2714** |
| 8 | 5.0762 | 5.2659 | 5.2836 | 5.3697 | 4.5379 | 5.6232 | 4.6957 | 5.0037 | **14.864** |
| 9 | 4.3217 | 4.0468 | 5.0562 | 4.0824 | 3.8712 | 5.2403 | 3.5320 | 5.1132 | **6.9366** |
| 10 | **5.8178** | 4.9864 | 4.6358 | 4.2858 | 3.3925 | 4.5989 | 3.8341 | 4.4569 | 5.2270 |
| 11 | 1.6884 | 5.6068 | 4.4474 | 4.2764 | 4.7479 | 5.4493 | 2.8042 | 4.4433 | **6.0664** |
| 12 | 1.9403 | 3.6799 | 4.5301 | 3.7105 | 4.5395 | 4.7260 | 2.5238 | 4.7219 | **6.7804** |

**Table 3. UCIQE of evaluation results on test datasets.**

| Dataset | UDCP | DRDCP | Fusion | UNTV | MLFcGAN | FUnIEGAN | UWCNN | WaterNet | Ours |
|---------|------|-------|--------|------|---------|----------|-------|----------|------|
| *UIEBD* | 0.5977 | 0.5722 | 0.5210 | 0.6304 | 0.5319 | 0.5590 | 0.5014 | 0.5328 | **0.6710** |
| *EUVP* | 0.6030 | 0.6122 | 0.5383 | 0.6203 | 0.5564 | 0.5878 | 0.5286 | 0.5707 | **0.6635** |
| *RUIE* | 0.5462 | 0.5307 | 0.4470 | 0.6071 | 0.4505 | 0.5044 | 0.4386 | 0.4469 | **0.6552** |

**Table 4. UIQM of evaluation results on test datasets.**

| Dataset | UDCP | DRDCP | Fusion | UNTV | MLFcGAN | FUnIEGAN | UWCNN | WaterNet | Ours |
|---------|------|-------|--------|------|---------|----------|-------|----------|------|
| *UIEBD* | 5.8751 | 4.5949 | 4.4241 | 3.4642 | 4.5593 | 4.5285 | 3.2181 | 4.4847 | **7.1265** |
| *EUVP* | 4.3669 | 4.5201 | 4.4614 | 4.5359 | 3.3719 | 4.9082 | 3.2465 | 4.6092 | **7.1927** |
| *RUIE* | 2.7712 | 4.8552 | 4.0543 | 4.2525 | 4.1490 | 4.7124 | 2.2367 | 4.1881 | **7.2856** |

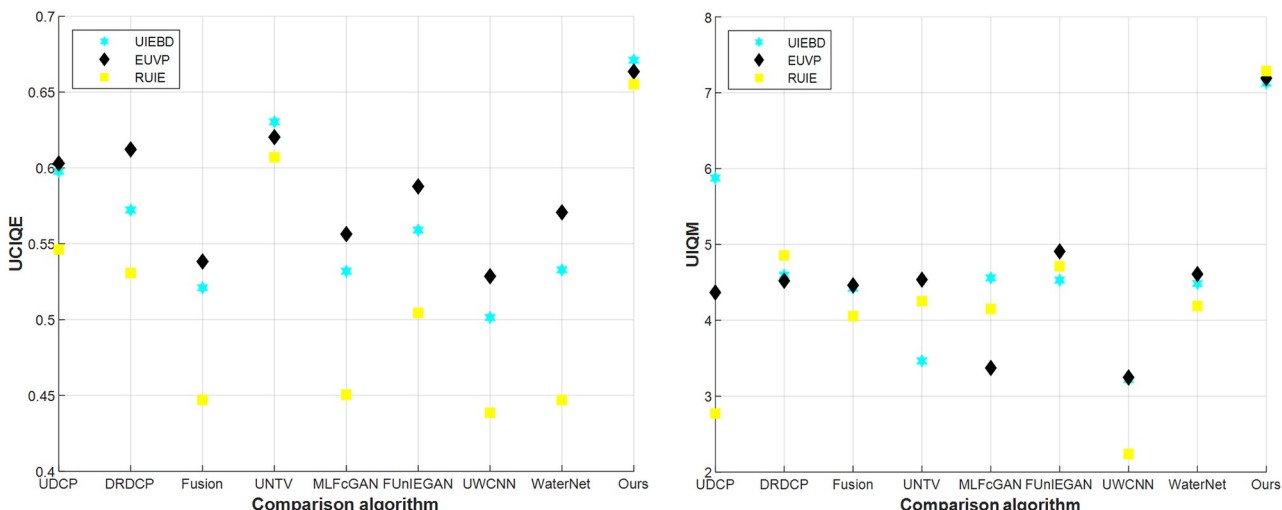

**Fig 10. UCIQE and UIQM of evaluation results on test datasets.**

**Table 5. Average running time (seconds).**

| Method | UDCP | DRDCP | Fusion | UNTV | MLFcGAN | FUnIEGAN | UWCNN | WaterNet | Ours |
|--------|------|-------|--------|------|---------|----------|-------|----------|------|
| *Average* | 0.181 | 0.237 | 1.172 | 2.428 | 1.424 | 1.826 | 1.701 | 8.301 | 1.351 |

## Complexity analysis

We compare the average time this code runs on three common datasets with non-deep learning and deep learning methods to analyze the complexity of the algorithm (CPU: Intel i7-6700HQ 2.60GHz; GPU: NVIDIA RTX 2070 8GB). Among them, non-deep learning algorithms include UDCP, DRDCP, Fusion, and UNTV. Deep learning algorithms include MLFcGAN, FUnIEGAN, UWCNN, and WaterNet. The running time of deep learning is only the test time, excluding the training time. The average processing time of each method is shown in Table 5.

We note that our method runs slower than the UDCP and DRDCP when only considering the non-deep learning methods. The reason is that we deal with the contrast, color, and

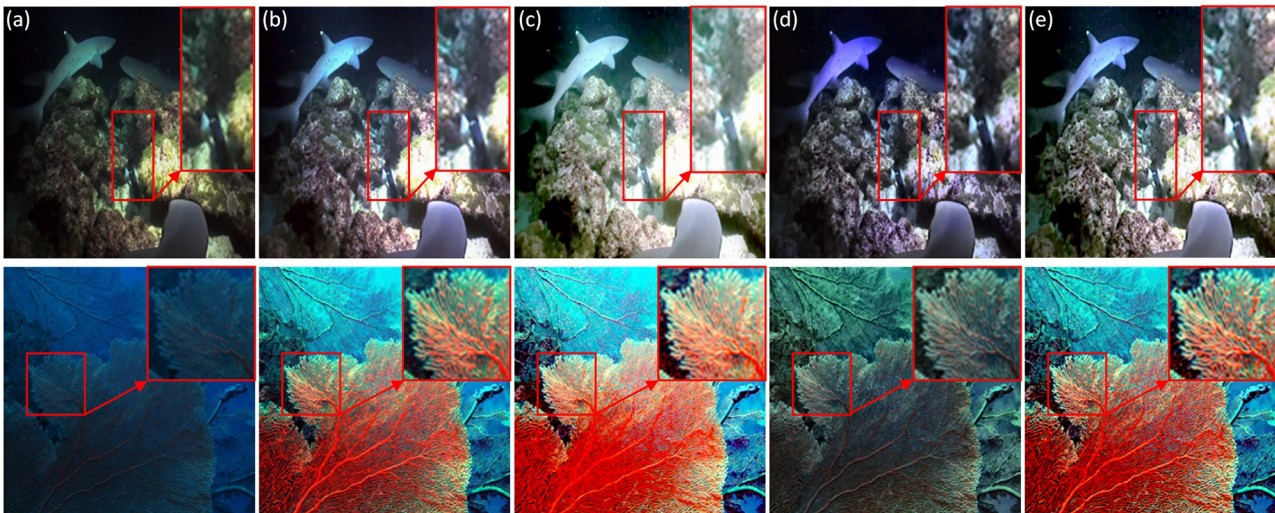

**Fig 11. Comparison of details of each input component.** (a) Degraded image; (b) First input image; (c) Second input image; (d) Third input image; (e) Ours.

dehazing of the underwater image separately, and the additional image enhancement effect increases the runtime. However, our running speed is better than that of Fusion and UNTV, which basically meets the requirements of underwater real-time performance. Deep learning algorithms not only take one to two days to train but also consume a lot of memory and test time is not as good as traditional algorithms. According to various subjective and objective evaluation metrics, our method outperforms other single restoration methods when dealing with complex underwater environments. Therefore, our work still has strong advantages in terms of comprehensive speed and enhancement effect.

## Ablation study

In this sub-section, the input image of each module component is compared with the fused clear image for detail information recovery experiment, as shown in Fig 11, the upper right corner is the local red mark area. The analysis shows that the input image can effectively complete contrast enhancement, color correction and dehazing respectively. Excessive color correction of the second input image will lead to serious exposure. The third input image can eliminate dense haze, but with distorted brightness and color. In contrast, our method establishes more generality than three input images that lack optimization capabilities.

After that, we verify that each input component is indispensable to the PSNR, UCIQE and UIQM evaluation metric. A denotes ours without contrast enhancement component, B represents ours without color correction component, C indicates ours without defogging component and D is the complete frame. The PSNR index is a full reference evaluation, which needs to be compared with the ground truth. Images with numerically higher PSNR values are generally considered better quality. However, there are no ground truth images in the RUIE dataset, so only the mean PSNR values on the UIEBD and EUVP datasets are compared. UCIQE and UIQM are non reference evaluation indicators, we compared the values of each component on the three data sets, as shown in Fig 12. We note that the PSNR, UCIQE and UIQM values of the unremoved input modules are all higher than other models, indicating that each module of input plays an important role.

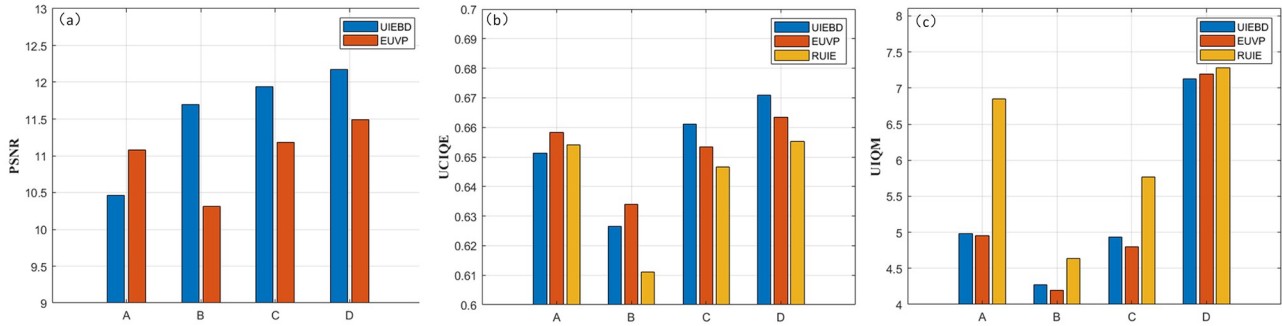

**Fig 12. PSNR, UCIQE and UIQM value of ablation experiment.** (a) PSNR; (b) UCIQE; (c) UIQM.

## Application test

In this sub-section, we adopt the performance of human target detection and saliency target detection based on YOLOv5 prediction algorithm to verify its application effect. The evaluation of human detection is measured by confidence, which the higher the value, the more influential the algorithm application. Saliency detection aims to identify the significant part containing helpful information in an underwater image. The more detailed information, the more precise the image, as shown in Fig 13.

We evaluate the performance of the algorithm through SURF [30] feature point matching and corner detection [31]. SURF test compares the matching number of feature points before and after the algorithm processing, judges the processing effect, and then evaluates the algorithm's performance. Generally speaking, the better the processing effect of the algorithm, the more the number of feature points matching. In terms of false and missed detection of corners, the method based on gray-scale images is better than that of original images directly. Therefore, the images in the experiments are converted into gray-scale, where the number of corner detections is directly proportional to the quality of the image. The experimental results are shown in Fig 14, tagged with the number of matching points and corners in the top left-hand.

Experimental results show that the proposed algorithm can match more feature points and the number of corner detection increases significantly. The degraded underwater image has no confidence value because it is low and will not be displayed. Where saliency target area is white, and the background area is black. It can be seen from the significant image that our

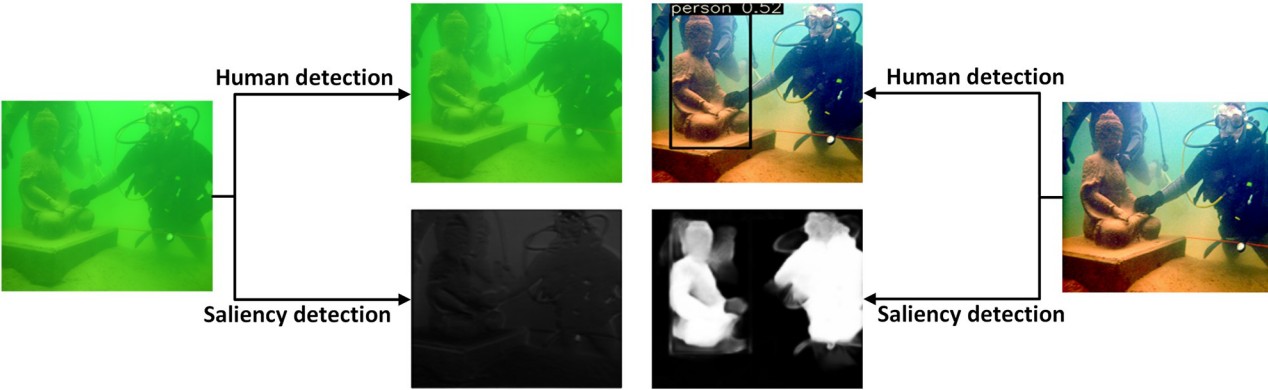

**Fig 13. Human and saliency target detection.**

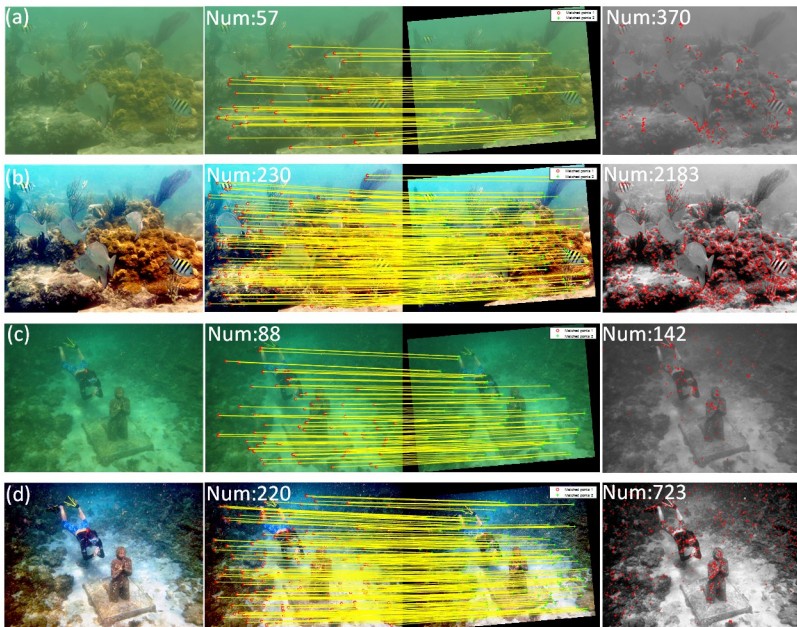

**Fig 14. SURF matching and corner detection.** (a) Degraded image 1, SURF and corner; (b) Corrected image 1, SURF and corner; (c) Degraded image 2, SURF and corner; (d) Corrected image 2, SURF and corner.

algorithm accurately recognized the region of interest, while the original image is not recognized.

Finally, in order to reflect the practical utilization of our work, our method is applied to underwater real video. The original video is decomposed into one frame at a frame rate of 30 frames per second, then each frame is fused and enhanced, and the corrected video is finally synthesized. As shown in Fig 15, two original images of three underwater videos are intercepted for experimental comparison. The real underwater video and enhanced video have been uploaded to Google Drive.

The experimental results show that the pretreatment effect of underwater video is clearer, and can reflect more abundant environmental information, which has practical application value.

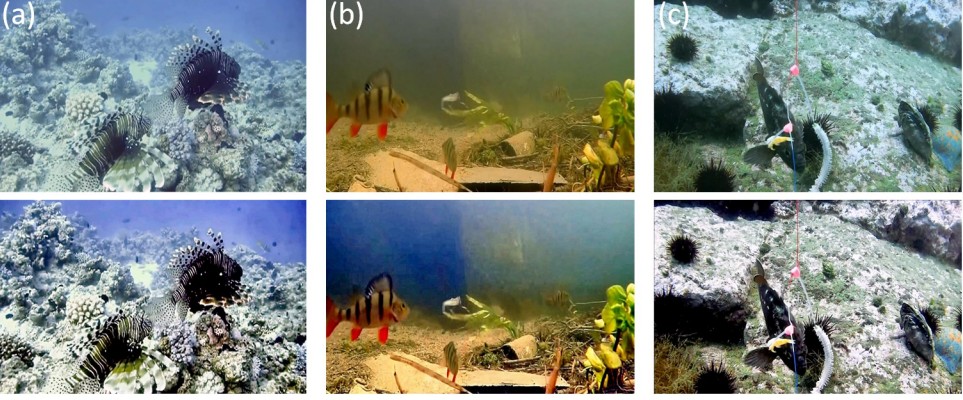

**Fig 15. Real underwater video test.** (a) Image from video 1; (b) Image from video 2; (c) Image from video 3.

## Discussion

Our method generally outperforms other contrasting algorithms, but still has certain limitations. For example, in UCIQE and UIQM indicators, the algorithm is not the optimal value. The background regions of the deeply blurred states in image 8 and image 10 in Fig 9 cannot be effectively recovered. In the processing of real underwater video, it is found that it cannot process low-resolution moving images. At the same time, according to the comparative running time, the complexity of the algorithm needs to be further improved.

## Conclusion

The complexity of the underwater imaging environment leads to low contrast, color distortion, blur, and other degradation problems within the acquired images. By targeting these issues, we propose an underwater optical image enhancement methodology based on the fusion of Retinex and transmittance optimization in this paper. Initially, by quantifying the gray values of each channel in the fuzzy underwater images, effective improvement in the contrast can be seen. Inspired by the Retinex model, dynamic adaptive stretch compensation is adopted to address image color deviation. The restored image obtained by the inverse double transmittance algorithm can deal with turbidity successfully. Next, the weights of the three input images calculated on different scales are obtained to reflect the basic features and necessary information of images. Finally, multi-scale pixel-level fusion is employed to construct the Laplacian pyramid and Gaussian pyramid for input and weight maps, resulting in a much more optimized underwater image. The experimental results indicate that our method can improve the image clarity and contrast while also correcting the color imbalance and retaining the image details.

## Author Contributions

**Conceptualization:** Tie Li, Tianfei Zhou.

**Data curation:** Tianfei Zhou.

**Formal analysis:** Tie Li, Tianfei Zhou.

**Investigation:** Tie Li.

**Methodology:** Tie Li.

**Project administration:** Tie Li.

**Resources:** Tianfei Zhou.

**Software:** Tianfei Zhou.

**Supervision:** Tie Li.

**Validation:** Tianfei Zhou.

**Visualization:** Tie Li.

**Writing – original draft:** Tie Li, Tianfei Zhou.

**Writing – review & editing:** Tie Li, Tianfei Zhou.

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
