## [Decision Letter · Decision Letter 0]

6 May 2022

PONE-D-22-07396A Novel Multi-scale Fusion Framework via Retinex and Transmittance Optimization for Underwater Sensing Scene Image EnhancementPLOS ONE

Dear Dr. Zhou,

Thank you for submitting your manuscript to PLOS ONE. After careful consideration, we feel that it has merit but does not fully meet PLOS ONE’s publication criteria as it currently stands. Therefore, we invite you to submit a revised version of the manuscript that addresses the points raised during the review process.

We look forward to receiving your revised manuscript.

Kind regards,

Sen Xiang

Academic Editor

PLOS ONE

Journal Requirements:

Reviewers' comments:

Reviewer's Responses to Questions

**Comments to the Author**

1. Is the manuscript technically sound, and do the data support the conclusions?

Reviewer #1: Partly

Reviewer #2: Yes

2. Has the statistical analysis been performed appropriately and rigorously? 

Reviewer #1: Yes

Reviewer #2: Yes

3. Have the authors made all data underlying the findings in their manuscript fully available?

Reviewer #1: Yes

Reviewer #2: Yes

4. Is the manuscript presented in an intelligible fashion and written in standard English?

Reviewer #1: No

Reviewer #2: No

5. Review Comments to the Author

Reviewer #1: Some comments should be concerned below:

C1. The proposed method should be proved in some real videos of underwater scenes.

C2. The runtime of the proposed method should be compared with some non deep learning and deep learning state-of-the-art methods on some public datasets.

C3. The mainline of Introduction should be clear so that readers can better understand their intentions.

C4. There are some formatting problems, such as basic alignment of all paragraphs.

C5. There is an error in Fig. 1 where the reflected light direction of suspended particles is not uniform.

Missing references:

[1]Underwater Image Enhancement via Medium Transmission-Guided Multi-Color Space Embedding. Chongyi Li, Saeed Anwar, Junhui Hou, Runmin Cong, Chunle Guo, Wenqi Ren. IEEE Transactions on Image Processing, 2021.

[2]Bayesian Retinex Underwater Image Enhancement. Peixian Zhuang, Chongyi Li, Jiamin Wu. Engineering Applications of Artificial Intelligence, 2021.

Reviewer #2: The paper consists of a series of image enhancement algorithms such as improved multi-scale Retinex with color preservation (IMSRCP), histogram quantization, the red channel prior is integrated to the total transmittance estimation, and extracting the weights of the preprocessed clear input images. However, I have a few questions:

-- most of the methods included in the overall framework of this method are already well-known. In the proposed methodology, some integrated methods are combined in sequence to get better human perception results. In this way, the manuscript holds novelty. However, the author must discuss their original contribution.

-- In section 2 Methodology, improved multi-scale Retinex with color preservation (IMSRCP), histogram quantization, the red channel prior is integrated to the total transmittance estimation, and extracting the weights of the preprocessed clear input images. But in the explanation of methodology in each sub-sections, the order of the methodology seems to be in correct. Reorder the methods used in the methodology based on the summary as briefed in the overview of the proposed methodology of Section 2.

-- In sub-section 2.2 Design weight diagrams, the normalized weight map of the k-th input image, K = 3. But in the Fig. 2, the decomposition level is 5. Explanations are needed because this introduces ambiguity.

-- the Proposed Methodology Fig. 2 needs a self-explanation with proper order of the methods used, and further, the Fig. 2 needs more artistic representation and check for the normalized weight map and its corresponding level of decomposition.

-- Some ablation studies need to be performed in the experiments to prove the effectiveness of each method. If possible PSNR validation can be carried out in ablation study.

-- the author must discuss failure cases of their method (if any).

-- there are few typo errors present in the manuscript, the author must correct these errors.

-- notations used in certain equations were not properly defined.

-- the english of the paper must be revised thoroughly as the reviewer has found numerous grammatical errors and punctuation mistakes.

-- the work is very interesting, however too many algorithms will lead to consume more time and computing power.

6. PLOS authors have the option to publish the peer review history of their article (what does this mean?). If published, this will include your full peer review and any attached files.

Reviewer #1: No

Reviewer #2: No

---

## [Author Response · Author response to Decision Letter 0]

20 Jun 2022

We have submitted the revised draft of "A Novel Multi-scale Fusion Framework via Retinex and Transmittance Optimization for Underwater Sensing Scene Image Enhancement"(ID: PONE-D-22-07396). Thank you for giving us the opportunity to modify it. We have carefully read your decision letter and made changes based on the comments of two reviewers, the revised content is in the 'Response to Reviewers',which we wish to be considered for publication in PLOS ONE.

---

## [Decision Letter · Decision Letter 1]

25 Jul 2022

PONE-D-22-07396R1A Novel Multi-scale Fusion Framework via Retinex and Transmittance Optimization for Underwater Sensing Scene Image EnhancementPLOS ONE

Dear Dr. Zhou,

Thank you for submitting your manuscript to PLOS ONE. After careful consideration, we feel that it has merit but does not fully meet PLOS ONE’s publication criteria as it currently stands. Therefore, we invite you to submit a revised version of the manuscript that addresses the points raised during the review process.

We look forward to receiving your revised manuscript.

Kind regards,

Sen Xiang

Academic Editor

PLOS ONE

Journal Requirements:

Reviewers' comments:

Reviewer's Responses to Questions

**Comments to the Author**

1. If the authors have adequately addressed your comments raised in a previous round of review and you feel that this manuscript is now acceptable for publication, you may indicate that here to bypass the “Comments to the Author” section, enter your conflict of interest statement in the “Confidential to Editor” section, and submit your "Accept" recommendation.

Reviewer #1: All comments have been addressed

Reviewer #2: All comments have been addressed

2. Is the manuscript technically sound, and do the data support the conclusions?

Reviewer #1: Yes

Reviewer #2: Yes

3. Has the statistical analysis been performed appropriately and rigorously? 

Reviewer #1: Yes

Reviewer #2: Yes

4. Have the authors made all data underlying the findings in their manuscript fully available?

Reviewer #1: Yes

Reviewer #2: Yes

5. Is the manuscript presented in an intelligible fashion and written in standard English?

Reviewer #1: Yes

Reviewer #2: No

6. Review Comments to the Author

Reviewer #1: We have no comments on this manuscript, and recommend the authors to check the full text carefully. This manuscript presented an underwater image enhancement method based on a fusion of Retinex and transmittance optimization. This method quantified the gray values of each channel to improved the contrast, used the dynamic adaptive stretch compensation to solve the color deviation, and obtained the restored image through an inverse double transmittance algorithm. Multi-scale pixel-level fusion is used to construct the Laplacian and Gaussian pyramids for input and weight maps, resulting in the final underwater image.

Reviewer #2: 1. The paper title should be concise and as short as possible, and Include keywords (refer - as per Authors Guidelines of this journal)

2. I appreciate that the authors has carried out the Ablation study based on PSNR metric and why not other metrics such as UCIQE and UIQM metrics. Further, the authors can also include Ablation study with input image and its corresponding output. This can provide the readers to understand the variations in the output image as well as the importance of each block of components as proposed in Fig. 2.

3. Make sure all figures are as per the journal dpi format, since some of the images are seems to be of less resolution (refer - as per Authors Guidelines of this journal).

4. English grammar correction (such as punctuation, typo mistakes) needs to be done.

7. PLOS authors have the option to publish the peer review history of their article (what does this mean?). If published, this will include your full peer review and any attached files.

Reviewer #1: No

Reviewer #2: No

---

## [Author Response · Author response to Decision Letter 1]

26 Aug 2022

We have submitted the revised draft of 'Multi-scale Fusion Framework via Retinex and Transmittance Optimization for Underwater Image Enhancement'(ID: PONE-D-22-07396). Thank you for giving us the opportunity to modify it. We have carefully read your decision letter and made changes based on the comments of two reviewers, the revised content is in the 'Response to Reviewers',which we wish to be considered for publication in PLOS ONE.

---

## [Decision Letter · Decision Letter 2]

12 Sep 2022

Multi-scale fusion framework via retinex and transmittance optimization for underwater image enhancement

PONE-D-22-07396R2

Dear Dr. Zhou,

We’re pleased to inform you that your manuscript has been judged scientifically suitable for publication and will be formally accepted for publication once it meets all outstanding technical requirements.

Kind regards,

Sen Xiang

Academic Editor

PLOS ONE

Additional Editor Comments (optional):

Reviewers' comments:

Reviewer's Responses to Questions

**Comments to the Author**

1. If the authors have adequately addressed your comments raised in a previous round of review and you feel that this manuscript is now acceptable for publication, you may indicate that here to bypass the “Comments to the Author” section, enter your conflict of interest statement in the “Confidential to Editor” section, and submit your "Accept" recommendation.

Reviewer #1: All comments have been addressed

Reviewer #2: All comments have been addressed

2. Is the manuscript technically sound, and do the data support the conclusions?

Reviewer #1: Yes

Reviewer #2: Yes

3. Has the statistical analysis been performed appropriately and rigorously? 

Reviewer #1: Yes

Reviewer #2: Yes

4. Have the authors made all data underlying the findings in their manuscript fully available?

Reviewer #1: Yes

Reviewer #2: Yes

5. Is the manuscript presented in an intelligible fashion and written in standard English?

Reviewer #1: Yes

Reviewer #2: Yes

6. Review Comments to the Author

Reviewer #1: We have no comments on this manuscript. This method quantified the gray values of each channel to improve the contrast, used the dynamic adaptive stretch compensation to solve the color deviation, and obtained the restored image through an inverse double transmittance algorithm. Multiscale pixel-level fusion is used to construct the Laplacian and Gaussian pyramids for input and weight maps, resulting in the final underwater image.

Reviewer #2: The manuscript can be accepted for publication, as there is no further comments on this manuscript, and also I appreciate the authors for addressing all the comments.

7. PLOS authors have the option to publish the peer review history of their article (what does this mean?). If published, this will include your full peer review and any attached files.

Reviewer #1: No

Reviewer #2: No

---

## [Editor Report · Acceptance letter]

14 Sep 2022

PONE-D-22-07396R2 

Multi-scale fusion framework via retinex and transmittance optimization for underwater image enhancement 

Dear Dr. Zhou:

I'm pleased to inform you that your manuscript has been deemed suitable for publication in PLOS ONE. Congratulations! Your manuscript is now with our production department. 

Kind regards, 

on behalf of

Dr. Sen Xiang 

Academic Editor

PLOS ONE